# Your Pain Pleases Others: The Influence of Social Interaction Patterns and Group Identity on Schadenfreude

**DOI:** 10.3390/bs14020079

**Published:** 2024-01-23

**Authors:** Binghai Sun, Tongyin Huang, Ying Wu, Liting Fan

**Affiliations:** School of Psychology, Zhejiang Normal University, Jinhua 321004, China; jky18@zjnu.cn (B.S.); zqw0717@zjnu.edu.cn (T.H.)

**Keywords:** schadenfreude, competition, outgroup derogation, ingroup favoritism, disgust

## Abstract

Schadenfreude is a joyful emotional response to the misfortune of others. Individuals’ schadenfreude over the misfortunes of ingroup and outgroup members may vary depending on how groups interact. Accordingly, this study examines the effects of different social interaction patterns and group identity on schadenfreude and their mechanisms. The study participants were Chinese college students. Study 1 (*n* = 83) investigated whether there are differences in individuals’ schadenfreude towards ingroup and outgroup members under two different patterns of social interaction: cooperation and competition. On the basis of this study, Study 2 (*n* = 73) focused on the mechanisms underlying the influence of individuals’ Schadenfreude on ingroup and outgroup members in competitive situations, and the mediating role of disgust. It was found that there was an interaction between group identity and social interaction patterns, with people showing more schadenfreude over the misfortunes of outgroup members than ingroup members, and competitive situations increasing disgust and schadenfreude over outgroup members. However, no differences were found in individuals’ schadenfreude towards ingroup and outgroup members in cooperative situations. This is instructive in terms of real-life intergroup relations as well as patterns of social interaction. This proves that cooperation and group contact is a way to reduce schadenfreude.

## 1. Introduction

Schadenfreude is a complex psychological phenomenon that has been described as a disguised form of aggression that harms social relationships [1]. Researchers have become interested in schadenfreude recently because it is the antithesis of empathy [2]. The term “schadenfreude” is derived from German and means to take pleasure in the suffering of others [2,3]. Schadenfreude refers to an individual’s joyful emotional response to the misfortune of others [4]. Schadenfreude appears to have a positive emotional valence in the context of an individual’s emotional experience, but at the root of it, it is an antisocial sense that is enjoyed while others are being harmed [5]. Self-interest may be subjugated to schadenfreude, in which people revel in the misfortunes of their rivals even when doing so harms both the individuals involved and society in general [6]. A number of studies on schadenfreude have revealed that it enhances behavior that is antisocial [7,8]. According to empirical studies, people are more prone to behave impolitely online when they are feeling schadenfreude. This is because when individuals gloat over the misfortunes of others, they do not perceive online trolling behavior as problematic, nor do they realize that such behavior is frowned upon by others [8]. Another study revealed that kids as young as 4 years old already experience the emotion of schadenfreude and adopt strategies for avoidance when faced with their schadenfreude targets’ need for assistance [9].

On the basis of the definition of schadenfreude as “taking pleasure in the misfortune of others”, van Dijk added two points to make the definition of schadenfreude more explicit. On the one hand, the misfortune that occurs to the target of schadenfreude is caused by the environment or others rather than by the experience of schadenfreude, and if it is, it is likely an abuse, which is more harmful than what we typically perceive as schadenfreude. On the other hand, this pleasure is not obtained from winning the actual competition, as doing so would result in not only schadenfreude but also multiple emotions of satisfaction and pride in succeeding [10]. As a result, in this study, we closely adhered to these two points in the experiment.

At present, researchers mainly use the situational hypothesis paradigm to study schadenfreude. The situational hypothesis paradigm creates a scene in which others encounter unfortunate events, usually including unfortunate video, picture, or text descriptions [11], and allows participants to have an imaginary emotional experience under the set scene, and then measures the participants’ schadenfreude by the questionnaire method. Cikara asked participants to read about an unfortunate event about an opponent, and after reading each story, subjects were asked to answer the question—“How happy did this make you feel?”, on a scale of 0–100 [12].

Although it is commonly recognized that schadenfreude tends to be antisocial in nature, what factors specifically affect an individual’s schadenfreude? The factors influencing schadenfreude have attracted the attention of numerous studies, and they can be roughly separated into individual and group components. Among them, individual factors include self-esteem, empathy, depression, etc. [5,13] and group factors include social identity, patterns of social interaction and stereotypes, etc. [7,14].

In a variety of contests or events, inverted cheering is frequently seen [7]. For instance, in a football game, when the crowd-supported team wins and the opposing team displays loss, the crowd frequently boos or utters sarcastic remarks to indicate their joy. Even if the crowd is not a direct participant in the game, doing it this way will not promote future friendship between the two teams. In this instance, it is simple to recognize that a football game is a competition between two teams, which inherently involves the interaction and relationship between groups. Sometimes, individuals may enjoy more over the fact that the unfortunate person is a member of a certain group than the fact of their misfortune in and of itself. Numerous research projects have revealed that schadenfreude is influenced by group identity (e.g., ingroup and outgroup members) as well as patterns of intergroup social interaction [6,7]. Exploring the phenomena and mechanism of schadenfreude at the group level has practical significance and social usefulness since people’s group emotions and group behaviors are stronger in groups than when they are acting individually.

### 1.1. Intergroup Schadenfreude

Previous studies have revealed that the degree of schadenfreude toward the misfortunes of ingroup and outgroup members vary. This classification of identities is often based on a certain criterion or a certain characteristic, where individuals who jointly meet the same criteria are grouped together and given a common identity as a group.

In a number of studies, some researchers use groups from the real world to conduct the study directly [7,14], others may use certain criteria or activities to divide the participants into groups artificially [11,15]. Specifically, The Minimal Group Paradigm will be used for those who accept self-defined criteria, while those who adopt existing intergroup linkages will be evaluated based on factors like country, supported party, supported team, etc.

According to the social identity theory [7], individuals are more likely to adopt positive attitudes toward the group to which they belong when they have a sense of belonging and emotional support from that group. Social identity effectively provides an answer to the question, “Who are we”, because people frequently describe themselves in terms of the groups they are a part of [7]. Three fundamental psychological processes—social classification, social comparison, and positive differentiation—are involved in the formation of social identity [16,17]. Social classification refers to classifying different individuals, understanding the social environment by knowing the categories of belonging, and initiatively distinguishing others as ingroups or outgroups when classifying. When individuals are categorized, they deliberately place themselves in the inner group and adopt some inner group traits as personal traits, obfuscating the distinction between the self and the inner group [7]. People will compare themselves to others on an upward or downward slant in order to make a more thorough appraisal of oneself through such relative comparisons, according to Festinger’s social comparison theory [18]. Tajfel (1982) expanded social comparison from the individual level to the group level, arguing that when comparing and evaluating ingroups and outgroups, individuals will be more inclined to evaluate the ingroup in a positive attitude and the outgroup in a negative one [17]. Positive differentiation refers to the fact that individuals will actively display their strengths so that they can achieve self-esteem or self-motivation by performing better than outgroup members. Successful positive differentiation increases group identity as well as self-esteem of ingroup members and satisfies the individual’s quest for positive self-evaluation. Therefore, social identity theory indicates that when an individual perceives that he or she is in a particular social group, and at the same time recognizes that the group has special emotional significance and value for him or her, it makes the individual’s behavior clearly distinguishable from that of the group.

Ingroup favoritism and outgroup derogation are repeatedly mentioned in studies related to schadenfreude [15]. When an individual’s evaluation and interpretation of another person’s experience varies according to the identity of the group to which the other person belongs, the eventual group evaluation produces two different kinds of emotions, i.e., those directed toward the ingroup and those directed toward the outgroup, and in the process, ingroup preference and outgroup bias are created. It has been noted that the higher an individual’s identification with an ingroup, the lower the level of schadenfreude he or she shows for negative news about the ingroup and the higher the level of schadenfreude shown for negative news about the outgroup, and that the level of schadenfreude further positively predicts an individual’s willingness to communicate about negative news [14]. Similarly, Combs focused on the severity of unfortunate events and found that both economic problems of lesser magnitude and severe troop casualties and political damage elicited schadenfreude reactions from members of outgroups [6]. Individuals often empathize with their peers or teammates when they see them suffer misfortune, whereas empathy often turns to schadenfreude when they see their opponents or enemies suffer misfortune [19].

### 1.2. Patterns of Social Interaction and Schadenfreude

In previous studies of group member-related schadenfreude, researchers have explored a number of predictors of schadenfreude, focusing primarily on groups with a history of rivalry or hostile individuals or groups [20]. This shows that the patterns of social interaction is an important influence [21]. Patterns of social interaction between groups can often be characterized as cooperation and competition [22].

When an individual’s group is in a competitive or hostile relationship with the other, the misfortunes of the ingroup members cause people to empathize and feel sad and sorry for them, but the misfortunes of the rivals cause people to have pleasurable emotional reactions [23]. Since there is a dominant norm of group favoritism in group relations that compels group members to prioritize the protection of the interests of the ingroup before considering the interests of others [24], intergroup relations may provide a special breeding ground for schadenfreude [14]. Previous research has found that competitive relationships between groups can further increase intergroup conflict [25]. Placing two groups in an openly competitive relationship can usually enhance intergroup prejudice and hostility [26]. Even if the two groups do not engage in overt competition, there is an asymmetry in power and access to valuable resources for the different groups, which predicts perceptions of competitiveness [27]. Out of the importance attached to winning and losing outcomes, when groups are in a competitive situation with each other, they will show happiness for the competitor’s misfortunes and loss for the competitor’s gains [3].

Intergroup competitive interaction styles have been shown in different empirical studies to promote individual schadenfreude towards outgroups, However, the effect on schadenfreude when groups are in cooperative interaction styles has not been explored. Intergroup cooperation is the coordination of goals between at least two teams by recognizing their interdependence and the need to work together to contribute to the success of the mission. Allport’s (1954) study proposed that intergroup cooperation must include two elements, including shared processes (e.g., interacting together) and shared outcomes (e.g., a common destiny) [28]. The cooperative approach to intergroup interaction allows the two groups to share a common destiny, and the common destiny represents a relationship between the groups in which it is realized that the goals of the two groups are linked, and that ultimately the goals of the two groups can be achieved only if each member of the two groups achieves their goals. Numerous studies have shown that intergroup cooperation can be effective in reducing intergroup bias [28,29]. Miller suggested that intergroup cooperation leads to more individualized and positive impressions of outgroup members by ingroup members [29]. In addition, cooperative interactions can increase the evaluation of outgroup members, which can lead to more consistent and balanced attitudes and behaviors of individuals towards the outgroup while also enhancing the positive emotional experience of individuals [30].

### 1.3. The Role of Disgust

Disgust causes specific facial expressions, subjective experiences, physiological and behavioral responses, and is considered to be a basic human emotion [31]. Disgust originated from the rejection of putrid and unclean food, and the rejection of the object of disgust can avoid the ingestion of food that is harmful to the human body and serve as a protection for human survival. As human society develops, the variety of disgusting stimuli increases. Rozin’s proposed theoretical model of primitively constructed disgusting emotions categorizes disgust into four areas, namely animal reminder disgust, core disgust, moral disgust, and interpersonal disgust [32]. Subsequent research on disgust has shown that the emotion of disgust is social in nature, related to interpersonal prejudice, group prejudice, and caused by strangers or disliked people, i.e., interpersonal disgust [33]. The emotion of interpersonal disgust expresses the individual’s emotion of turning away from interpersonal or group relationships [34]. According to interdependence theory, there is a negative interdependence between groups when competing. This is manifested in an increased understanding of homogenization among the same groups, which can further increase prejudice against outgroups [35]. Empirical studies have also shown that competition increases favoritism toward ingroups and prejudice toward outgroups, which in turn tends to trigger individual aversion to competing groups [8].

Numerous studies on schadenfreude have shown that disgust is one of the common predictors that trigger schadenfreude [36]. People feel happier about the downfall of a disagreeable, rude object than the downfall of a likeable, friendly one [37]. Zhao used a situational hypothetical paradigm in which participants were asked to imagine an unfortunate event happening to a liked person and a disgusted person, respectively, and to assess their own emotions. In addition to having fMRI scans performed on the participants throughout the experiment, the results showed that individuals showed more sadness when misfortune happened to a liked person, and more indifference or even schadenfreude when misfortune happened to a disgusted person. Differences in activation of the left and right insula of individuals existed when imagining the misfortune of a liked versus a disgusted person. The degree of activation of an individual’s insula during an unfortunate encounter with a disgusted person can serve as one of important indicators of schadenfreude [38]. In the Powell and Smith study, participants were told in advance of learning about an unfortunate event that had happened to someone that the person had previously committed book plagiarism, which succeeded in triggering the participants’ disgust to the target, and it turned out that this disgusted person did indeed elicit more schadenfreude [39]. Although no study has directly pointed to disgust as a psychological mechanism for the effect of intergroup interactions on schadenfreude, through existing theories and existing empirical studies, we introduce the emotion of disgust in order to explore whether disgust plays a mediating role in it.

By analyzing the existing studies, we believe that the following issues need to be further clarified. First, when ingroup and outgroup members are in a cooperative mode of interaction, does it reduce the amount of schadenfreude that individuals show when members of the external group experience misfortune? Second, when ingroup and outgroup members are in competing modes of interaction, are the effects of ingroup and outgroup members identity on schadenfreude mediated through disgust? Third, previous studies have directly compared the differences in individuals’ displays of schadenfreude to ingroups and outgroups, but have failed to specify whether outgroup misfortunes increase individuals’ schadenfreude, ingroup misfortunes decrease individuals’ schadenfreude, or both [7].

Based on the above questions, 2 experiments were designed in this study. Experiment 1 examined how ingroup and outgroup members show schadenfreude over others’ misfortunes under different patterns of social interaction (cooperation, competition, independence). Experiment 2 intended to add control relationship groups to the competing interaction approach, further explored how ingroup and outgroup members identities influence schadenfreude over others’ unfortunate displays, and examined the role of disgust in this.

## 2. Experiment 1: The Effects of Social Interaction Patterns and Group Identity on Schadenfreude

The aim of Experiment 1 was to investigate whether there are differences in schadenfreude evoked by different patterns of social interaction and whether there are differences in schadenfreude toward ingroup and outgroup members. A mixed experimental design with 3 (patterns of social interaction: competition, cooperation, independence) × 2 group identity: ingroup members, outgroup members) was used in this experiment. The between-subjects variable was the patterns of social interaction situation, the within-subjects variable was group identity, and the dependent variable was schadenfreude. In this experiment, we hypothesized that schadenfreude in the independent situation would be significantly lower than in the competitive situation, significantly higher than in the cooperative situation, and that there would be more schadenfreude directed toward outgroup members than toward ingroup members.

### 2.1. Method

#### 2.1.1. Participants

Ninety non-psychology major Chinese college students between the age range of 18 to 26 years old participated in the experiment (female = 72, male = 18). Seven of the participants failed the tests of attention check and their data were not used for further data analysis. Thus, a total of eighty-three subjects were included in Experiment 1 into the further data analysis (females = 69, *M*_age_ = 21.90). At the end of the experiment, all participants were given a small amount of money as a reward.

#### 2.1.2. Materials

Negative events. On the basis of interviews, news collection and literature analysis, we compiled negative event materials about schadenfreude, evaluated the event severity and life closeness of negative event materials, and selected event materials with moderate degree to ensure the effectiveness of materials [38].

We focused on the universality of negative events during the collection process, which means that these negative events should be prevalent in daily life. We combined a total of 16 negative events, improved the event description, and managed the word count to ensure that each negative event had an equal word count. Negative events include “He/She sat on a chair with gum in it” “He/She was riding a bike when got a flat tire on a gravel road” and so on. Twenty-two graduate students in psychology (17 females; *M*_age_ = 23.91, *SD*_age_ = 1.41) rated this negative events. They rated each negative event on a 5-point scale for its severity (1 = very not serious, 5 = very serious) and its closeness to life (1 = not at all close to life, 5 = very close to life). According to the evaluation’s results, 16 negative events were suitable for use in formal experiments because of their moderate severity (*M* = 3.35, *SD* = 0.64, α = 0.91) and ideal closeness to life (*M* = 4.41, *SD* = 0.53, α = 0.91).

#### 2.1.3. Procedure

The experiment was designed and run using E-Prime 2.0.

The whole experiment consists of six parts (Figure 1A).

First, participants were divided into groups based on randomization after providing basic personal information (Figure 1B). Before each experiment, we needed to ensure that six participants who did not know each other participated in the experiment at the same time. The participants were informed that the collection of their basic information would serve as the foundation for the ensuing experimental groupings. In effect, we randomly assigned participants to either the red or blue team, forming two three-person teams.

Second, we manipulated patterns of social interaction. Two teams were randomly assigned to either a competitive situation, a cooperative situation, or an independent situation (the two teams were independent of each other and did not interfere with each other). In the competitive situation, the two groups of participants were told that the next rapid response keying tasks to be performed was a competition between the teams. The red and blue teams were competitors, and the team with the higher overall score would win the prize. In the cooperative situation, the two groups of participants were told that the next rapid response keying tasks would require both teams to work together. If both teams reached a total of 120 points within 3 min, both teams could jointly receive a reward, if not, neither team could receive a reward. In addition, in the independent situation, participants were told that if their team scored a total of 60 points within 3 min, then their team would be able to receive a reward.

Third, participants assigned to different patterns of social interaction were asked to complete a rapid response keying task (Figure 1C) [40]. In the rapid response keying tasks, a master pattern (standard stimulus) appears on the screen and participants compared the color and shape of the master pattern with that of the subsequent pattern (comparison stimulus). If the subsequent pattern is a different color from the main pattern, press “L”; if it is a different shape, press “F”; and if neither the color nor the shape is the same, press the “space bar”. For each correct answer, one point would be added to the team’s total score, and for each incorrect answer, one point would be deducted from the team’s total score. Once participants had no questions about the rules, there would be a practice phase followed by a 3 min formal task.

Fourth, participants completed tests of attention check. After completing the material reading scoring task, participants were asked to respond to the following three questions, “Which team are you on?” “What other team is playing?” “What is the relationship between your team and the other team?” Participants who answered all three questions incorrectly were considered to have failed the attention check, and their data were not included in subsequent data analyses.

Fifth, participants reported the levels of schadenfreude provoked by unfortunate events about ingroup members and outgroup members (Figure 1D). After completing the rapid response keying tasks, participants were presented with 16 negative events. Eight of these negative events occurred with ingroup members, and the other eight events occurred with outgroup members. Then, participants were asked to assess their feelings, i.e., “How happy did this make you feel? “(a slider bar: 0 = Not happy at all, 100 = Very happy) [12,20]. The average of that score was used as an indicator of schadenfreude.

Sixth, we eliminated the negative effects that this experiment may bring to the participants. All participants were paid an equal amount of money and were informed of the true purpose of this experiment.

### 2.2. Results and Discussion

A repeated-measures ANOVA in SPSS 26.0 was conducted with the patterns of social interaction (cooperation, competition, independence) as the between-subjects variable, the target of schadenfreude (ingroup, outgroup) as the within-subjects variable, participant’s schadenfreude as the dependent variable, and gender as the control variable. The results showed a significant main effect of the target of schadenfreude, *F* (1, 79) = 6.59, *p* = 0.012, *η_p_^2^* = 0.08. Participants reported more schadenfreude when negative events occurred with outgroup members (*M* = 155.81, *SD* = 140.82) compared to ingroup members (*M* = 94.92, *SD* = 82.04). The main effect of the patterns of social interaction was not significant, *F* (2, 79) = 0.91, *p* = 0.407, *η_p_*^2^ = 0.02. According to participants’ reports of schadenfreude, there were no significant differences in competitive situation (*M* = 159.15, *SD* = 112.56), cooperative situation (*M* = 102.95, *SD* = 87.54), and independent situation (*M* = 109.92, *SD* = 102.77). The interaction between patterns of social interaction and the target of schadenfreude was significant, *F* (2, 79) = 4.91, *p* = 0.010, *η_p_^2^* = 0.11. Further simple effects analyses revealed that participants reported having more schadenfreude to outgroup members (*M* = 215.53, *SD* = 163.22) than to the ingroup members (*M* = 102.77, *SD* = 81.12) in the competitive situation, *F* (1, 79) = 35.94, *p* < 0.001, *η_p_*^2^ = 0.31. There was no significant difference in participants’ schadenfreude toward ingroup and outgroup members in either the cooperative situation [*M* _ingroup_ = 86.43, *SD* = 74.37; *M* _outgroup_ = 119.46, *SD* = 107.29, *F* (1, 79) = 3.83, *p* = 0.054, *η_p_*^2^ = 0.05] or independent situation [*M* _ingroup_ = 95.00, *SD* = 93.08; *M* _outgroup_ = 124.84, *SD* = 124.95; *F* (1, 79) = 3.60, *p* = 0.062, *η_p_*^2^ = 0.04]. Figure 2 shows the result clearly.

The results of Experiment 1 showed that negative events for outgroup members evoked more schadenfreude from participants than for ingroup members. But furthermore, the intergroup effect was only observed in competitive situation and not in cooperative and independent situations. Based on this study, Experiment 2 would focus on competitive situation to explore possible psychological mechanisms underlying the intergroup effect of schadenfreude. In addition, there was a further unexplained problem with the results of Experiment 1, namely, whether the intergroup differences in schadenfreude were due to the role of outgroup derogation or ingroup favoritism? Experiment 2 added a control group to the variable of targets of schadenfreude to further examine the effect of group membership on schadenfreude.

## 3. Experiment 2: The Effect of Group Identity on Schadenfreude in Competitive Contexts: The Role of Disgust

The purpose of Experiment 2 was to explore whether there would be a difference in schadenfreude directed to the ingroup and outgroup members in the situation of group competition, and whether this difference was more due to ingroup favoritism or outgroup derogation, in addition to wanting to explore the mediating role of disgust. Experiment 2 used a within-subjects design with three conditions (group identity: ingroup members, outgroup members, control group members). Consistent with Experiment 1, the dependent variable was schadenfreude. In Experiment 2, we hypothesized, first, that participants would display schadenfreude significantly more toward outgroup members than toward ingroup members and control group members in competitive situation. Second, the participants were increasing schadenfreude through disgust for outgroup members.

### 3.1. Method

#### 3.1.1. Participants

Seventy-eight non-psychology major Chinese college students between the age range of 18 to 25 years old participated in the experiment (female = 60, male = 18). Five of the participants failed the tests of attention check and their data were not used for further data analysis. Thus, a total of seventy-three subjects were included in Experiment 2 into the further data analysis (females = 57, M_age_ = 22.30). Again, participants who participated in the study were paid a small amount of money as payment at the end of the experiment.

#### 3.1.2. Materials

Negative events. Consistent with Experiment 1.

#### 3.1.3. Procedure

The whole experiment consists of six parts (Figure 3).

Experiment 2 maintained the same process as Experiment 1, with only the following three differences.

First, Experiment 2 retained only the competitive situation. Because the results of Experiment 1 observed an intergroup effect of schadenfreude only in the competitive situation, in order to further clarify how this intergroup effect develops, Experiment 2 no longer examined schadenfreude in the cooperative situation as well as in the independent situation.

Second, to examine participants’ disgust to ingroup and outgroup members, after the attention check, participants were asked to answer two questions, “How much do you disgust the team you are on? “ “How much do you disgust the opposing team?” (slider bars: 0 = not at all, 100 = very disgust).

Third, among the targets of schadenfreude, we added the control group, i.e., the outgroup members who had nothing to do with the rapid response keying tasks (Figure 3). That is, participants were presented with five unfortunate events for ingroup members, five unfortunate events for outgroup members, and five unfortunate events for control group members.

### 3.2. Results and Discussion

Evoking schadenfreude by different targets. A one-way analysis of variance (ANOVA) was conducted with the target of schadenfreude (outgroup, ingroup, control group) as the within-subjects variable and the degree of schadenfreude after numerical transformation as the dependent variable (In the homogeneity of variance test, it was concluded that the variance was not homogeneity, *p* = 0.036 < 0.05. Therefore, we transformed the variable “schadenfreude” and then it passed the homogeneity of variance test, *p* = 0.110). The results showed that there were significant differences in individuals’ schadenfreude toward members of different target of schadenfreude, *F* (2, 215) = 8.31, *p* < 0.001, *η_p_*^2^ = 0.07. Multiple comparisons after LSD post hoc indicated that participants were significantly more schadenfreude in the outgroup members (*M* _outgroup_ = 204.78, *SD* = 127.78) than ingroup members (*M* _ingroup_ = 111.04, *SD* = 100.91, *p* < 0.001) and control group members (*M* _control group_ = 136.17, *SD* = 113.25, *p* = 0.003. At the same time, there was no significant difference between participants’ levels of schadenfreude for members of the ingroup and those for control group members, *p* = 0.363.

Analysis of the mediating role of disgust. Kendall’s tau-b analysis and Pearson’s analysis were used to analyze the correlations between the target of schadenfreude, disgust, and schadenfreude. The results showed that the target of schadenfreude (0 = ingroup; 1 = outgroup) showed significant positive correlation with both disgust (*r* = 0.16, *p* = 0.023) and schadenfreude (*r* = 0.30, *p* < 0.001), and a significant positive correlation between disgust and schadenfreude (*r* = 0.23, *p* = 0.005). The PROCESS statistical plug-in developed by Hayes (2013) was used to conduct the mediation effects test. Among them, the target of schadenfreude (0 = ingroup; 1 = outgroup) was the independent variable, disgust was the mediator variable, and schadenfreude was the dependent variable. The total, the direct, and the indirect effects of the target on schadenfreude were shown to be significant in Model 4 of PROCESS (none of the upper or lower bounds of its bootstrap 95% confidence intervals contained zero). The results showed that there was a significant correlation between group identity and disgust (*p* < 0.05), *β* = 8.78, 95%*CI* (1.09,16.47). There was a significant correlation between disgust and schadenfreude (*p* < 0.001), *β* = 1.35, 95%, *CI* (0.59, 2.11); Group identity was significantly correlated with schadenfreude (*p* < 0.001), *β* = 81.76, 95%*CI* (46.14, 117.37) (Table 1). Under the condition of competition, the indirect effect of group identity on schadenfreude through aversion is 11.87, and the mediating effect accounts for 12.68% of the total effect. In other words, under the condition of competition, the mediating effect between group identification and schadenfreude is significant, the mediating model is established, and the influence of some intermediary group identification on schadenfreude is disgusted (Figure 4).

Experiment 2 explored the effect of the target of schadenfreude on schadenfreude, and the mediating role of disgust in the pattern of competitive. The results of the experiment showed that, first, participants were significantly more schadenfreude towards outgroup members than control and ingroup members. Second, participants’ schadenfreude toward ingroup members were not significantly different from the control group members. Third, under the intergroup competition condition, participants’ disgust towards the outgroup increased, which increased the degree of schadenfreude toward the outgroup.

## 4. General Discussion

Schadenfreude is a common social emotion, and people’s emotional responses to what happens to others are influenced by their social environment [41]. The purpose of this study is to explore the effects of the patterns of social interaction and group identity on schadenfreude, as well as the role that disgust plays in this process.

### 4.1. Outgroup Depreciation and Ingroup Preference

The results of both Experiments 1 and 2 showed that participants experienced significantly more schadenfreude with outgroup members than with ingroup members. In this study, the participants in the experiment were college students who did not know or interact with each other prior to the experiment, as opposed to members of real groups which are interdependent and have deep emotions. The participants were allowed to fill out basic personal information before the experiment began and were made to believe that the groups were divided according to similar personal information, and after completing the experiment, it was unlikely that the participants would interact with each other again. And the experiment was conducted only once, and there were no previous hostile or mutually helpful relationships, and there was no emotional antagonism between the groups. Nonetheless, individuals respond emotionally to ingroup and outgroup members differently depending on the group they are in. It has been found in existing research that even when individuals are categorized in a minimalist group manner, the group identity assigned to them by that group causes individuals to prefer ingroup members and have positive attitudes and emotions toward them, while outgroup members show prejudice and have negative attitudes and emotions, which is referred to as ingroup preference and outgroup depreciation. Tajfel (1982) argues that whenever individuals have the opportunity to engage in a kind of identity segmentation, they are likely to engage in biased behaviors, and this process is social categorization [17]. Group differentiation and discrimination, ingroup favoritism, etc. are common group behaviors, and individuals produce group behaviors through two main processes: social categorization and social comparison [42].

From a theoretical perspective, Tajfel’s proposed Social Identity Theory and Smith’s Intergroup Emotion Theory can also explain the results obtained in this study from different perspectives [17,43]. According to Tajfel’s social identity theory, an individual’s self-concept derives in part from the perception of the social group to which the individual belongs, which is referred to as the collective self or social identity. In addition, people strive for a positive social identity and want the group to which they belong to be distinguished from other groups by significant advantages. When news of misfortune for other groups is perceived as favorable to people striving for positive social recognition, individuals perceive their own group as gaining an advantage, i.e., they feel a pleasant emotion about the event. Smith’s theory of intergroup affect suggests that the strength of people’s identity with the ingroup, and in particular the affective component of ingroup identification, is crucial, and that the strength of people’s affective identification within the ingroup is a key determinant of outgroup schadenfreude. Emotions at the group level differ from those at the individual level in that group emotions depend on the level of group identity and group emotions pervade the entire group. Group emotions are more influential and stable than individual-level emotions [3].

The degree of social identity affects the emergence of group behaviors, such as discrimination and devaluation of outgroups and solidarity and motivation of ingroups [43]. One study used a situational hypothesis paradigm to explore service industry personnel’s display of schadenfreude toward high-achieving outgroup members [44]. The results of the study suggested that as individuals perceive themselves as belonging to the service industry to a greater extent, they are more likely to experience strong schadenfreude when they see misfortunes befalling government officials from the outgroup. The more an individual socially identifies with the group to which he or she belongs, the more likely he or she is to schadenfreude at the misfortunes of the outgroup. This group role is even more pronounced in the interactive mode of intergroup competition, just as fans of the Boston Red Sox and New York Yankees, two rival baseball teams, show pleasure when they see their opponents score very few runs [45].

The results of Experiment 1 and Experiment 2 are consistent with both theories proposed by previous authors [15,44]. Individuals will carry a derogatory attitude toward outgroup members and show more positive attitudes and more favoritism toward ingroup members, thus schadenfreude is significantly more directed toward outgroup members than ingroup members.

### 4.2. Intergroup Competition Increases Outgroup Schadenfreude: The Role of Disgust

Among the many factors that influence intergroup schadenfreude, the pattern of social interaction has been recognized as one of the most important variables [46]. Placing two groups in a competitive situation generally increases prejudice and hostility towards outgroup members [47].

The effects of the patterns of social interaction on schadenfreude were explored in Experiment 1, which showed that participants in competitive condition were significantly more schadenfreude than those in the cooperative and independent conditions. Individuals were found to display more schadenfreude in competition condition between intergroup, suggesting that the more hostile the group’s patterns of social interaction, the greater the degree of schadenfreude toward outgroup members, in line with previous findings. Schadenfreude occurs under competitive conditions. When two groups are in a competitive or antagonistic relationship, unfortunate events for members of the ingroup cause individuals to empathize and feel pain, but the unfortunate events of a competitor tend to cause individuals to feel pleasant emotions [23].

In addition, whether high levels of schadenfreude under intergroup competitive condition are due to the roles of the ingroup and outgroup members or to the pattern of competition, and why individuals schadenfreude at higher levels for outgroup members than for ingroup members in competitive situation need to be further explored. This hypothesis was then re-tested in Experiment 2 by adding control group members as a baseline group. With the help of the control group as the baseline group, the results of the study showed that individuals in the group competition situation experienced significantly higher levels of schadenfreude than the control group, i.e., the group competition condition increased the level of schadenfreude of individuals towards out-group members. The results of the experiment proved the experimental hypothesis, that is, under group competition, individuals experience higher levels of schadenfreude when confronted with an unfortunate event that befalls an outgroup member compared to an ingroup member.

Numerous studies on the antecedents of schadenfreude have shown that disgust is the most prevalent predictor [7]. Regarding the factors investigated in the current study, if we disgust someone, we are more likely to derive pleasure from his or her misfortune [36,37]. Therefore a measure of disgust was introduced in Experiment 2 to explore the role that disgust plays in the effects of group identity on schadenfreude under the intergroup competition condition. The results of the study showed that disgust partially mediated the effect of group identity on schadenfreude under the group competition condition, i.e., under the group competition condition, individuals would increase their level of schadenfreude by increasing their level of disgust towards members of the outgroup when they were confronted with them.

### 4.3. Schadenfreude under Intergroup Cooperation: Long-Term Effects of Teamwork

In Experiment 1, the results of the study showed that there was no significant difference in the degree of individual schadenfreude between the conditions of group cooperation and group independence. There are two possible reasons for this result.

First, the group in which the experiment was conducted in this study was divided into subjects according to some similar basic personal information, and the members of the group did not know each other before the experiment, and their relationship with each other belonged to the temporary being established, so that the social distance between individuals was far away from each other. Social distance refers to the degree to which an individual perceives the closeness or distance of the relationship between him/herself and others [48], and defines the social distance between an individual and another person mainly from the perspectives of familiarity, intimacy, and similarity, i.e., the further the individual perceives the social distance from another person, the more unfamiliar and estranged is the relationship between the two persons. Although the subjects in this study were told that they were divided into groups by similar basic information and also collaborated with others to accomplish the task, familiarity and closeness between the subjects did not increase, so the degree of schadenfreude of individuals in the group cooperation condition was not significantly different from that of the control group.

Second, this study explored intergroup cooperation but neglected the long-term effects of teamwork. The degree of threat the environment poses to each group, the perceived functional benefit of the groups working together or separately, and the length of cooperation may all influence whether groups are better off interacting with each other than never interacting.

The two teams were asked to work together during the task, but after the experiment was over, it is likely that the participants would no longer engage in back-and-forth with each other, and the cohesive effect that would have been increased by teamwork would have diminished. The study found that maintaining different, but interrelated, work tasks was more conducive to reducing an individual’s group bias than when groups worked together on the same collaborative tasks [49]. Long-term teamwork interactions can shift members’ cognitive representations from “I” and “he” to a more inclusive “we”. However, in this study, the skills and time and effort required by each participant to participate in the cooperative task were relatively equal, the two groups brought almost the same contribution to the group task, and the members of the team worked together for a relatively short period of time, so the level of schadenfreude in the group cooperation condition was not significantly different from the control group.

### 4.4. Application Insights

Schadenfreude is a very common emotion that is often observed in real life. Schadenfreude, although considered a dark emotion, is not without its positive aspects. On an individual level, an unfortunate event of another person that triggers an individual to schadenfreude can protect an individual’s self-esteem and enhance an individual’s self-evaluation. For the group, schadenfreude about the outgroup can serve to unite the ingroup or exclude dissenters, but it can also enhance prejudice against the outgroup. When individuals perceive pleasure at the misfortune of others as a reward for themselves, individuals are likely to artificially inflict misfortune on others, and in extreme cases may commit atrocities against others, and when the other person is a member of an outgroup, the individual’s aggression is likely to be further reinforced [45]. We therefore believe that schadenfreude is not conducive to the development of intergroup harmony. Intergroup contact theory suggests that good group contact reduces intergroup prejudice. A good group contact is one that meets the requirements of equal status, shared positive goals, good cooperation, and clear social support for the groups in which the two parties are in contact. This provides a practical basis for guiding inter-individual and intergroup interactions in social life, which can include the following two points:

First, individuals should correctly understand and treat the emotion of schadenfreude. Schadenfreude is a common emotion that occurs in both interpersonal and group interactions. Individuals will not only gloat over the misfortunes of members of the outgroup, but also over the misfortunes of members of the ingroup, and do not need to fall into moral self-sanctions because of the emotion of schadenfreude. Individuals should learn to face up to such emotions, and if they are strong, they should channel them in time, and regard schadenfreude as an early warning of interpersonal and group social conflicts. We can optimize our interpersonal relationships by strengthening cooperation with others, increasing contact with others, reducing the feeling of schadenfreude to others, and truly empathizing with others.

Secondly, group contact and co-operation should be strengthened to reduce intergroup schadenfreude. In group relationships, individuals avoid information about the outgroup, resulting in a lack of contact and understanding between groups. In addition, there is often tangible or intangible competition between groups. These are prone to cause intergroup schadenfreude, which triggers malicious attack behaviors, intensifying intergroup conflicts and escalating conflicts. This is of guiding significance to the construction of group relations in the organization. Creating more opportunities for contact, interaction and cooperation through good group contact is conducive to easing intergroup prejudice and, to a certain extent, can dilute social conflicts such as interpersonal and group conflicts, avoiding the destruction of interpersonal and inter-group relations and the occurrence of unfortunate incidents. This helps the organization to progress and develop better.

### 4.5. Theoretical Implications

First, most of the previous studies have explored individual schadenfreude about group members based on objective groups in reality or by having subjects imagine themselves in a group. Some studies use objective real social groups as participants, but this cannot rule out additional variables, such as cultural differences and identity. The existence of empathy bias among groups is a recognized phenomenon, and the existing stereotypes of groups should also be controlled in the research. And the ecological validity of the approach of letting subjects imagine themselves belonging to a certain group is low. Therefore, the study created a truly equal and neutral group. Before the formal experiment, the participants were told to randomly assign them to the new group according to the similarity of their basic information, so as to ensure that the participants did not have any bias and stereotype between the ingroup and outgroup before the group assignment, and the group characteristics (such as familiarity, conflict history) could also be controlled in the experiment.

Second, most previous studies have explored schadenfreude under conditions of group competition, but have not explored other modes of interaction; this study enriches the research on the effect of modes of interaction of schadenfreude by adding the interactive mode of group cooperation and exploring its effect on schadenfreude.

Third, this study once again verified the theory of intergroup contact. Group bias is likely to lead to schadenfreude among outgroups, while contact between individuals and outgroups under favorable conditions can effectively reduce inter-group bias, and the optimal intergroup contact mainly includes the following four key conditions: Contact with outgroups in an atmosphere of equal status, share common goals, group cooperation, and authoritative legal support. Some empirical studies have shown that intergroup contact can eliminate prejudice or conflict between groups and improve attitudes towards out-groups. Swart (2007) conducted a longitudinal study of intergroup contact in conflict societies in South Africa, showing that intergroup contact can enhance positive attitudes among groups despite the adverse context of group conflict, in populations with high levels of group conflict, including whites, blacks, Indians, and people of color. Compared with groups in lower status, after intergroup contact with groups in higher status, individuals show more obvious changes in their attitudes towards out-groups [50,51]. Good intergroup contact can effectively reduce group prejudice, reduce the cultural differences between groups, weaken the extreme identification of the internal group, improve the evaluation and likability of the external group, and reduce the emotional reaction of schadenfreude of the external group.

Finally, the reasons for the differences in individuals’ schadenfreude towards ingroup and outgroup members in a intergroup competitive interaction style remain unclear. The introduction of the variable disgust in this study provides empirical support for exploring the factors that account for the occurrence of increased schadenfreude toward outgroup members under competitive conditions. In this study, based on previous research, we added outgroup members of non-competitive/cooperative groups as a baseline group, verified that group competition significantly increased outgroup members’ schadenfreude, and measured individuals’ disgust to outgroup members, confirming that the increase in individuals’ schadenfreude to outgroup members was due to an increase in disgust to outgroups.

### 4.6. Limitations and Directions for Future Research

One limitation of this study is that the emotion of schadenfreude is highly implicit, and the participants have a social approval effect when they report their schadenfreude subjectively. In future studies, the implicit measurement of schadenfreude can be used to exclude the social approval effect, and physiological indicators such as EEG and skin electricity can be used to provide a more objective basis for research.

The experimental group in this study was grouped according to some similar basic personal information, and the group members did not know each other before the experiment, and the relationship between them was temporary, and the individuals were relatively far away from each other. In the future, the effects of competition and cooperation on schadenfreude at different social distances between members (e.g., friends and strangers) can be explored.

In this study, a truly equal and neutral group was created to ensure that participants did not have any biases and stereotypes between the ingroup and outgroup before group assignment, and variables such as group characteristics could also be controlled in the experiment. However, in the real world, individuals between groups of strangers who see each other experiencing one of these events (such as a flat tire) may trigger a prosocial help that interferes with the emotional experience associated with schadenfreude, and future studies can continue to investigate whether it can trigger prosocial help and affect schadenfreude.

The results of this study indicated that there was no significant difference in participants’ schadenfreude toward ingroup and outgroup members in the pattern of social interaction in group cooperation. In this study, cooperation was operationalized as cooperation in which both parties have equal abilities and mutual benefits. Future research could explore different types of cooperation, such as exploring cooperation in which both parties have unequal abilities and mutual benefits, cooperation in which one party is weak and the other is strong, or cooperation in which one party profits more and the other less, and thus explore whether schadenfreude produces change.

## Figures and Tables

**Figure 1 behavsci-14-00079-f001:**
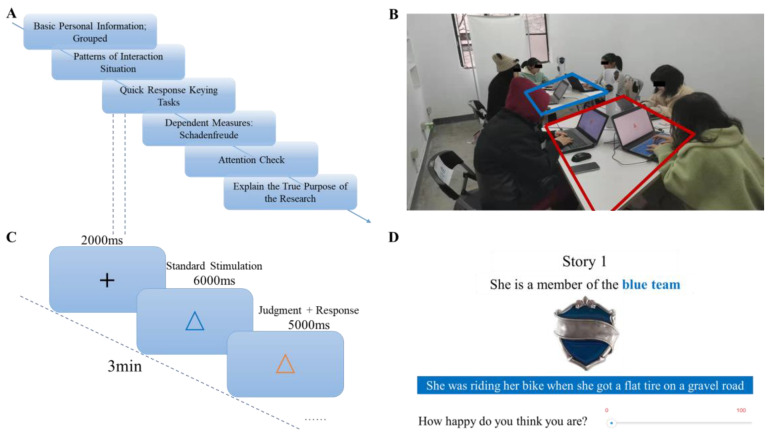
Experimental procedure and its details for Experimental 1. (**A**) Experimental flow chart. There are six parts to the experiment. (**B**) Experimental scenario setting: two groups of participants perform experimental tasks together. (**C**) Rapid response keying tasks. First, participants were shown a black “+” fixation point for 2000 milliseconds on a screen, and then a target shape (e.g., △) for 6000 milliseconds. Finally, a tone and another shape appeared on the screen, which participants had to judge within 5000 milliseconds: the shape of the shape changed, and the F key was pressed; The color changes, press J key; The color and shape are different press the space bar. (**D**) Measurement of schadenfreude in the event of misfortune to in-group or out-group members.

**Figure 2 behavsci-14-00079-f002:**
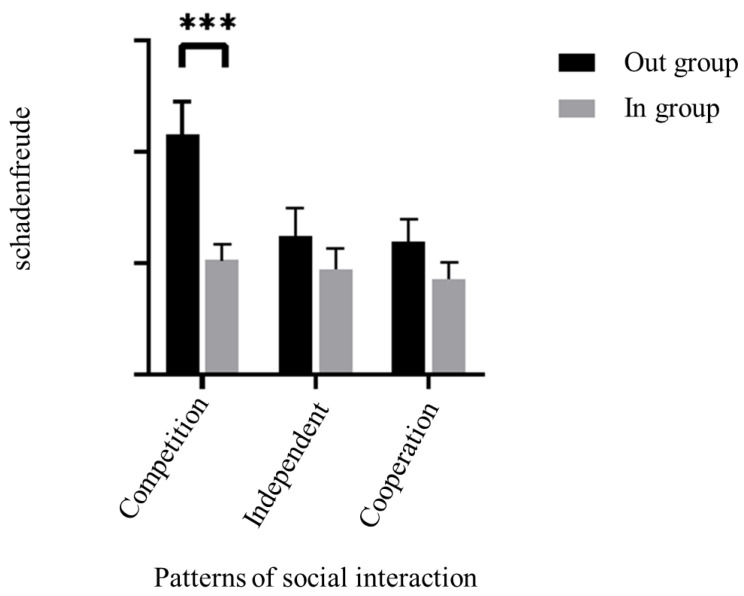
Differences in intergroup schadenfreude among different patterns of social interaction. (Note: *** *p* < 0.0001).

**Figure 3 behavsci-14-00079-f003:**
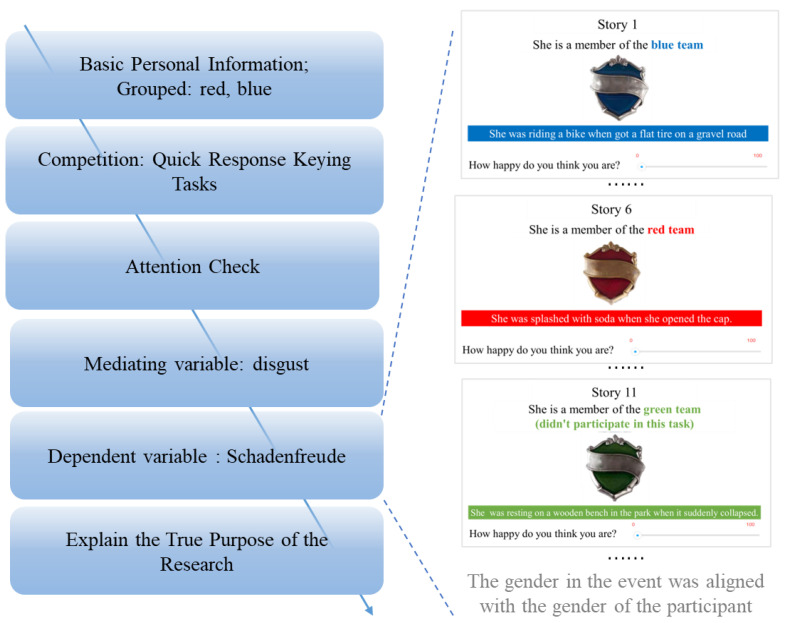
Experimental procedure and its details for Experimental 2.

**Figure 4 behavsci-14-00079-f004:**
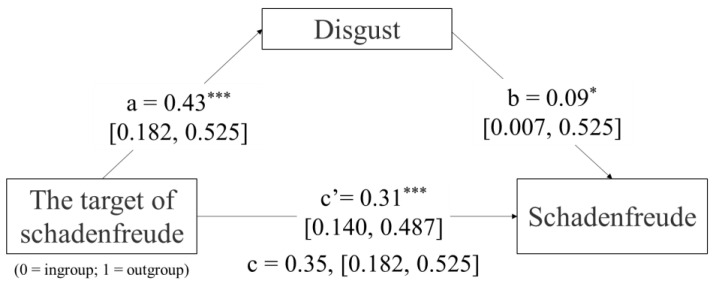
Mediation model of processes linking the target of schadenfreude and schadenfreude, mediated by disgust. (Note: * *p* < 0.05, *** *p* < 0.001).

**Table 1 behavsci-14-00079-t001:** In the competitive condition, disgust in group identity and schadenfreude mediated effect path analysis.

	Model Pathways	Effect Size	*p*	95% CI
Direct effect	Group identification—Schadenfreude	81.76	0.000	[46.14, 117.37]
Indirect effect	Group identification—Disgust—Schadenfreude	11.87		[0.60, 26.24]
Total effect		93.63	0.000	[57.25, 130.00]

## Data Availability

The data presented in this study are available on request from the corresponding author.

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
