# Peer review of "Your Pain Pleases Others: The Influence of Social Interaction Patterns and Group Identity on Schadenfreude"

_behavsci, 2024, doi:10.3390/bs14020079_

Round 1

Reviewer 1 Report

Comments and Suggestions for Authors

This is an interesting paper which does seem to extend our understanding of intergroup schadenfreude and how this relates to social interaction.  I could see that the results justified the conclusions and that the study was completed in a reliable and controlled manner. It is also useful to see a pattern of results which are consistent with established theoretical principles and the paper as a whole was interesting to read.  Thank you to the authors for submitting this for consideration and giving me the opportunity to read it. 

I do however have some questions about; certain aspects of the method, the decision to use a "minimal group" rather than a more ecologically valid set of group memberships and with the choice of a single disgust measure rather than the rating of a range of different emotions. I suspect that the first  elements can be discussed and justified relatively straightforwardly.  However the disgust aspect may require a bit more consideration. 

1) In terms of the methods, the sample sizes did seem adequate and the presence of significant findings would suggest that the studies were sufficiently powered.  However given the reputation that social psychology in particular has when it comes to transparency and replicability, I was surprised that the authors were not explicit about any involvement in Open Science.  I would expect to see details of a power analysis having been conducted.  This would establish not just that the authors had a big enough sample size but also that the sample size was overly large. The effect sizes are small in many of the ANOVA results so it would be useful to see these in context of the overall study power. It would also be useful to know whether the protocol was pre-registered.  The specificity of the analytic approach would suggest that this may in fact have been done - in which case the authors should be reporting this as it is would be a clear example of good practice. If not, it would be worth mentioning this in the limitations and exploring this for future work.  If the authors are "just getting started with" open science, there is a really accessible podcast from Brian Nosek here which might give them some ideas:  How ‘open science’ is changing psychological research, with Brian Nosek, PhD (apa.org)

2) My second point about the methods concerns the "severity" ratings for the 16 events included. From the examples provided these would indeed seem to be moderate severity - however it was not clear whether the authors were deliberately going for "moderate severity" events or whether the mean scores were simply an artefact of the way people use 5 point Likert scales.  We know that people are often reluctant to use the ends of the scale on a Likert measure so this may simply be due to "default" responding.  To help address this, the authors could explain more about the procedure for excluding the events which did not fit at both extremes of the scale.  For example it would be helpful here to understand whether any of the events were rejected for being too highly severe or whether there was simply a cut off of anything which was low in severity.  Where was the cut off?  Therefore more information on how these decisions were taken would be useful.  I would also suggest completing some form of statistical comparison between those accepted and rejected for both severity and representativeness of everyday life.  Perhaps t-tests comparing the retained vs rejected items for each variable? This is because although the representative of daily life results seem more convincing, the process should also be explained with the same level of detail as for the severity ones.  

3) The minimal group paradigm is long established and may be considered a conservative test of the hypotheses.  However there is a question over its ecological validity - even with the social interaction task acting as a social identity induction.  While vignette and minimal group studies are widely used, it would be worth recognising that this does not necessarily reflect real world responding. For example, a minimal group induction in the real world might be insufficient to produce the schadenfreude described here.  Alternatively it might be that viewing someone experiencing one of these events (e.g. the flat tire) would prompt a prosocial offer of assistance which would interfere with the experience of the emotions associated with schadenfreude. While these real world reactions are more variable and harder to predict, greater justification for the choice of a minimal group paradigm is needed at the outset.  It would be ideal if this could be justified ahead of existing alternative procedures such as using pre-existing social group memberships and then choosing a control group for experiment 2 from a different domain. And it would be even better if this could justify that the social interaction is a sufficient social identity induction to produce these effects. 

Similarly reflection on the strengths and weaknesses of this approach in the discussion needs to be stronger. 

4) The method of experiment 2 suggests that the disgust aspect was addressed through 2 self-report items and that the authors only enquired about disgust.  Am I correct in this?  If not, then greater clarity on this aspect is needed in the method.  If I am, the decision to single out and ask for ratings of a specific emotional reaction in such an overt way seems to me to promote rather than prevent socially desirable responding.  How have the authors prevented socially desirable responding and if they have not, what implications does this have for disgust as a mediating variable? I feel this is something which needs to be addressed but that this might be less straightforward than the previous three points and it is for this reason that I have given the overall recommendation I have. Given the importance of this to the authors' hypotheses, I would want to see a convincing level of reassurance on this final point in particular. 

Comments on the Quality of English Language

Generally the English language was clear and I know you are not writing in your first language so I commend you on doing this to your current standard. 

However there is a little work to be done on ensuring you have participant/verb agreement. For example, making sure that when you are speaking about more than one study, you refer to "empirical studies" and then use a plural form of the verb (e.g. "are" or "were" rather than "is" or "was").  This is something I would suggest checking through the manuscript as a whole. 

There are also places where you have referenced an external source but these references have come up as errors.  The first is on line 248.  These seem to be confined to the method sections of both experiments which may make them easier to locate. 

Finally the opening section of your introduction where you pose the rhetorical question as an attempt to engage and align the author and the reader is...speaking frankly...awkward.  The rest of your manuscript shows that you as an authoring team are more comfortable when writing in a factual way, so I would consider replacing this opening with a more academic style.  Normally I would not comment on this but I found myself feeling very awkward and slightly uncomfortable on your behalf and this is not a great "first impression" for your reader.  I would definitely consider revising. 

Author Response

Thank you very much for taking the time to review this manuscript. Thank you for your valuable suggestions for our manuscript, we have carefully read your suggestions and made revisions. Please see the attachment. If you have any questions, please do not hesitate to contact us. Thank you again for your valuable suggestions on our articles and for making our manuscripts even better.

Reviewer 2 Report

Comments and Suggestions for Authors

Title: consider shortening to less than 16 words

Abstract: should provide more Method details - eg size of sample, culture of sample etc - final sentences does not add anything new to the paper

Introduction

1. line 33-34 sentence needs revision

2. Good introduction but have not expanded on the Schadenfreude definition and its measurement - and the same for the concept of disgust

3. For example, there is a difference between dislike of a person and disgust with that person or their behaviour

Experiment 1

Participants - need course of study and cultural background

Materials - negative events to elicit Schadenfreude need to be validated??? Where else have they been used etc

Procedure 

1. This and later sections contain error messages that need to be resolved [first of many examples - line 248 

2. A simple measure of personal happiness is not a measure of Schadenfreude

3. Need to reference the statistical package used 

Results - well presented and understandable

Experiment 2

English expression [lines 363-5] does not make sense - this study appears not be completed in English and therefore the questions should be cited in their original language/form along with a better translation

Method - similar concerns as experiment 1

1. not enough information re participants

2. Schadenfreude does not equal [general] happiness 

3. Disgust is more than not liking a person

4. No experimental manipulation of disgust

General discussion

1. Good summary

2. Good theoretical discussion

3. Conclusions are consistent if the validity of the measures is accepted. For example discussion mentions disgust and loathing of a person, this has not been demonstrated in this study

4. Discussion mentions manipulation of social distance, needs more detail regarding this in the Method section

Application insights - authors conclude not conducive to intergroup harmony but

1. no applied solution to interpersonal behaviour

2. no applied/practical advice on increasing cooperation within competitive situations [eg what is good for the organisation to have competitive groups] 

Theoretical implications

1, laboratory research is not the end goal of research, it is the beginning and the end goal should be field or natural experiments [with variables such as age, gender, culture, abilities, etc

2. Again, no solutions to the problems associated with Schadenfreude

3. Agreed, the causes are still to be unexplained

Limitations and future research - is too brief, some of the issues raised above need to be addressed

References - OK

Comments on the Quality of English Language

There is a significant number of sentences that require reworking - should get a native English speaker to read this work

However, resolving this will not resolve other compelling issues

Author Response

(The authors gave the same response as above.)

Round 2

Reviewer 1 Report

Comments and Suggestions for Authors

Thank you for improving the manuscript in such a timely manner and for clearly highlighting the changes you made.  I am satisfied that this meets my comments - good luck with your future work!

Author Response

Thank you very much for your valuable comments and suggestions on our manuscript. Making changes in accordance with your comments allows us to progress our manuscript. Finally, thank you for your recognition of our manuscript. Thank you for replying to us, I wish you a happy life and smooth work!

Reviewer 2 Report

Comments and Suggestions for Authors

Most of the concerns have been addressed

Good practice to include a separate document highlighting the concerns of a reviewer and a note from the authors indicating how they addressed these concerns

There are some minor errors in this document - needs to be proofed again - for example under "Theoretical implications" there are two paragraphs numbered "Third"

Comments on the Quality of English Language

English is greatly improved

Author Response

Thank you very much for your valuable comments and suggestions on our manuscript. Making changes in accordance with your comments allows us to progress our manuscript. Thank you for reminding us that we found some minor errors in the document and re-proofread it. Finally, thank you again for your advice and help to us, thank you for your encouragement and recognition of us. I wish you a smooth and happy life and work in the future!